# Appraisal of Remote Sensing Technology for Groundwater Resource Management Perspective in Indus Basin

**Gulraiz Akhter** [1,2,*], **Yonggang Ge** [3,4,*], **Naveed Iqbal** [5], **Yanjun Shang** [6,7,8] **and Muhammad Hasan** [6,7,8]

1   China-Pakistan Joint Research Center on Earth Sciences, CAS-HEC, Islamabad 45320, Pakistan
2   Department of Earth Sciences, Quaid-i-Azam University, Islamabad 45320, Pakistan
3   Institute of Mountain Hazards and Environment, Chinese Academy of Sciences, Chengdu 610041, China
4   Key Laboratory of Mountain Hazards and Earth Surface Processes, Chinese Academy of Sciences, Chengdu 610041, China
5   Pakistan Council of Research in Water Resources, Islamabad 44790, Pakistan; naveed_spacian@yahoo.com
6   Key Laboratory of Shale Gas and Geoengineering, Institute of Geology and Geophysics, Chinese Academy of Sciences, Beijing 100029, China; jun94@mail.iggcas.ac.cn (Y.S.); hasan.mjiinnww@gmail.com (M.H.)
7   Institutions of Earth Science, Chinese Academy of Sciences, Beijing 100029, China
8   University of Chinese Academy of Sciences, Beijing 100049, China
*   Correspondence: agulraiz@qau.edu.pk (G.A.); gyg@imde.ac.cn (Y.G.)

**Abstract:** The dynamic nature and unsustainable exploitation of groundwater aquifers pose a range of management challenges. The accurate basin-wide hydrological assessment is very critical for the quantification of abstraction rates, spatial patterns of groundwater usage, recharge and discharge processes, and identification of critical areas having groundwater mining. This study provides the appraisal of remote sensing technology in comparison with traditionally prevailing tools and methodologies and introduces the practical use of remote sensing technology to bridge the data gaps. It demonstrates the example of Gravity Recovery and Climate Experiment (GRACE) satellite inferred Total Water Storage (TWS) information to quantify the behavior of the Upper Indus Plain Aquifer. The spatio-temporal changes in aquifer usage are investigated particularly for irrigation and anthropogenic purposes in general. The GRACE satellite is effective in capturing the water balance components. The basin-wide monthly scale groundwater storage monitoring is a big opportunity for groundwater managers and policymakers. The remote sensing integrated algorithms are useful tools to provide timely and valuable information on aquifer behavior. Such tools are potentially helpful to support the implementation of groundwater management strategies, especially in the developing world where data scarcity is a major challenge. Groundwater resources have not grown to meet the growing demands of the population, consequently, overexploitation of groundwater resources has occurred in these decades, leading to groundwater decline. However, future developments in the field of space technology are envisioned to overcome the currently faced spatio-temporal challenges.

**Keywords:** groundwater; remote sensing; Indus Basin; Pakistan; GRACE

## 1. Introduction

Groundwater is an underground finite resource contributing to agricultural maintenance and ecosystem sustainability. It acts as a buffer in droughts and helps in maintaining the water supplies in countries like Pakistan where surface water is more prone to climatic implications in addition to storage limitations. In Pakistan, groundwater fulfills approximately 90% of drinking water requirements and more than 60% of irrigation water supplies [1]. The Indus Basin is the largest basin in Pakistan [2] and serves as the main source of groundwater. More than one million tube wells are pumping fresh groundwater in the Upper Indus Plain—Punjab Province [3]. As a result, the water table is depleting and the water quality deteriorating [4,5]. In recent years, groundwater availability for irrigation has dropped from 5000 $m^3$ per capita to less than 1000 $m^3$ [6] due to population growth and agricultural land expansion.

Being an underground resource, the measurement, monitoring, and management of groundwater are more crucial than that of surface water. The set of tools in practice such as piezometric observations, groundwater modeling, and isotopic applications has inherent data, as well as scope, limitations in terms of spatio-temporal domains [7]. The credible and readily available information is a major concern for groundwater managers, which poses more pressure on the research community to strive for the development of new tools and algorithms. It is a fact that remote sensing techniques and GIS are very useful tools in the analysis of drainage networks, a study of surface morphological features, and their correlation with groundwater management prospects at the basin level. Before GIS and remote sensing technologies, identification of drainage systems within basins or sub-basins was attained using conventional approaches and topographic maps [8–10]. Nowadays, latest technologies are effectively utilized to develop significantly accurate pictures of several watershed delineation [11].

Due to input data constraints in terms of their spatial scales and temporal frequency, the existing groundwater modeling efforts are limited to only case study scales [12–14] and do not provide a holistic approach at the Indus Basin scale. On the other hand, the manual data collection through the piezometric ground observational network restricts the data availability up to biannual frequency. The manual data collection, done monthly through a network of around 2500–3000 piezometers in the Punjab province, again poses a big challenge. Therefore, the piezometric data is only available biannually (pre-monsoon and post-monsoon). Sustainable groundwater management demands controlled pumping, adoption of water conservation practices, and rainwater harvesting techniques. Groundwater monitoring is an essential component of any groundwater regulatory framework to ensure the proper implementation of groundwater policies [15]. Thus, an effective groundwater monitoring mechanism envisages establishing the success of groundwater regulatory policy. This fact drives the motivation to explore the alternate options and formulation of new algorithms based on the latest technological developments.

The increasing population, urbanization, enhanced agricultural productivity, and climate change are key factors that have hampered the balance between water demand and supply [16,17]. In this environment, it is felt among the scientific community that the challenges related to water resources are now more complex than before. The solutions demand the integration of interdisciplinary techniques such as hydrological modeling, remote sensing, geophysics, etc. Recently, the remote sensing datasets in the form of various precipitation products, digital elevation model (DEM), and land use/landcover are dominantly used for the development of hydrological models.

For the same reason, hydrologists either use remote sensing datasets directly as input in the models or for the calibration and validation of modeling results [7,18]. With the advancements in the field of space technology, multiple satellite missions are in space for the collection of valuable information on the various components of the water cycle. The global coverage with free and frequent data availability at fine spatial scales are the key features of remote sensing technology. The available remote sensing products offer a wide range of applications in hydrological studies. The recent advances in the field of satellite remote sensing (SRS) technology, such as satellite gravimetry, altimetry, and interferometry (soil moisture and land subsidence) have empowered the scientific community to investigate the solution of complex problems in a multi-dimensional frame.

NASA's Gravity Recovery and Climate Experiment (GRACE) is a unique satellite with the ability to capture all processes of the water cycle [7]. It provides information about the water changes, either stored on the surface of the earth or below. It is a unique satellite that senses the complete water cycle in one go. The successful applications of GRACE in various parts of the globe [19–27] have increased the interest and confidence of both the research community and the groundwater managers for its practical adaptability. GRACE-induced groundwater storage information is particularly useful for the assessment of variations in groundwater storage that characterize the aquifer behavior.

The objective of this study is to review the existing challenges of groundwater management in the Indus Basin, as a case study of the data-scarce region of the developing world as well as to evaluate the potential of satellite gravimetry as an opportunity for effective groundwater resource management. Moreover, it will highlight the merit and demerit of GRACE satellite-based adoptability at operational groundwater management scales.

## 2. Study Area

In the Indus Basin, the groundwater aquifer mainly exists in the Indus Plain, which falls both in Pakistan and India with moderate topographic variations. The part of the Indus Plain Aquifer in Pakistan is subdivided into Upper and Lower Indus Plain Aquifers. Upper Indus Plain Aquifer (UIPA) consists of four hydrological units, locally known as "doabs" in the Punjab province of Pakistan. These doabs (Thal, Chaj, Rechna, and Bari) represent the floodplains bounded by rivers (Figure 1). The total area of UIPA is about 107,704 km$^2$, which is highly fertile and productive agricultural land. The study area is part of the Indus Basin Irrigation System (IBIS), which is comprised of a network of canals to provide surface water for irrigation supplies. The UIPA is a well-transmissive and continuous unconfined aquifer where lithology varies from coarse sand to sandy loam with clay lenses [7]. The isotopic analysis shows that UIPA receives major recharge from rivers with contribution from the irrigation system and return flow from agricultural fields. The precipitation-induced recharge is mainly available in the central areas of the doabs [28]. Based on the forty-year records, the average annual rainfall (at Lahore, Multan, Faisalabad, Sialkot, and Jhelum) in the Punjab province is about 580 mm [29].

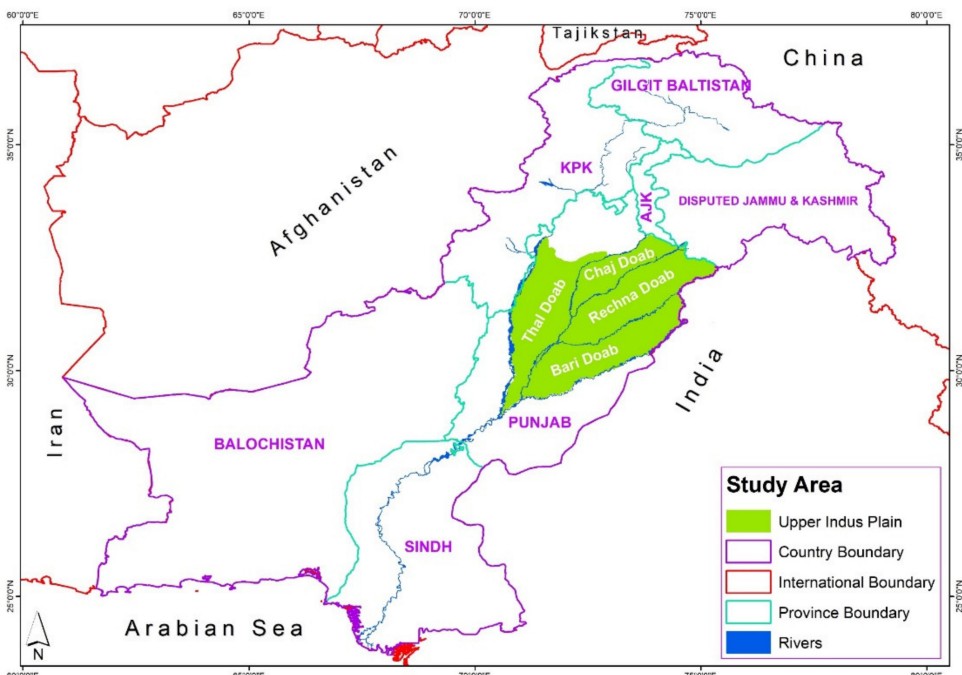

**Figure 1.** Location map of the study area.

The farmers are under pressure to increase productivity to meet the increasing demand for food and other agricultural supplies. As a result, farmers are drilling more tube wells to manage surface water shortage through groundwater pumping. Over time, it has not only resulted in an increased area under cultivation but also increased cropping intensity [30]. In the absence of groundwater regulation, farmers are independent to install any number of tube wells and at any depth, wherever they like. Consequently, the tube-wells growth has increased exponentially over time with more than one million tube wells only in the Punjab province (Figure 2). From these tube wells, about 85% are diesel operated, and the remaining 15% are electric. Over the last 20 years, electric tube wells have increased

from 12% in 2000 to 16% in 2018 [31]. The analysis of water level data reflects that the percentage of areas under deep depth to water table has increased about 20% (1991–2011), representing the fact of over-pumping [32]. Moreover, a further change of about 15% has been reported in terms of increase in area under deep (>6 m) water table depth (Figure 3). At present, about 50% of the total cultivated area in Punjab has a groundwater table depth below 6 m [33]. Due to low flows in the eastern transboundary river (Ravi and Sutlej) controlled by India, the parts of the aquifer falling in Bari and Rechna doabs (Figure 1) are under more stress.

The decreasing trend after 2016 is possibly due to a non-significant increase in further tube-wells growth as well as redundancy of old tube wells leading to becoming non-operational. It may be correlated with the fact of high tube-wells density achieved in the Punjab province. Out of 1.2 million, about 85% tube wells are only in the Punjab province, which facilitates about 90% of total groundwater abstraction in Pakistan [33]. This continuous tube-wells growth has led to an increase in cropping intensity from 80% in 1950 to 189% in 2017 [33]. Moreover, there is no subsidy on electricity, rather, the government is promoting the use of high-efficiency irrigation systems (HEIS) to reduce over-abstraction and wastage of precious groundwater [34].

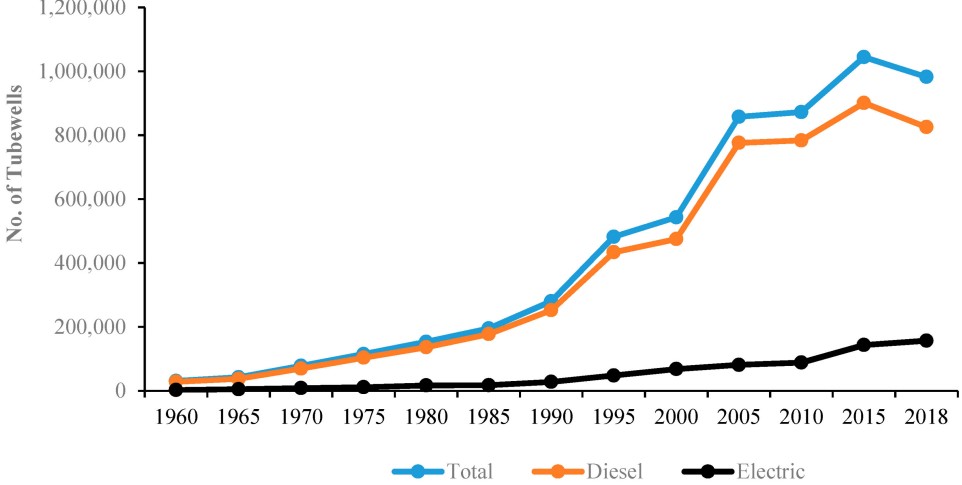

**Figure 2.** Tube-wells growth over time in the Punjab province (adopted from [33]).

The analysis of fluctuations in depth to water table shows that the significant depletion is reported in either urban areas like big cities (Lahore, Rawalpindi, Faisalabad, etc.) or the areas of tail-ends of canals or doabs (lower part of Bari Doab). In urban areas, the major reason for groundwater depletion is urbanization, which has restricted groundwater recharge. However, the over-abstraction of groundwater for anthropogenic purposes has caused an imbalance between recharge and pumping [7]. Like groundwater depletion, the groundwater recharge varies in spatio-temporal domains. In the Indus Basin, the major groundwater is received through seepage from the irrigation system [29]. Various studies have estimated groundwater recharge and reported groundwater depletion in lower parts of doab areas due to higher pumping than recharge [12–14]. Ref. [28] estimated that the lower parts of Bari (Lodhran, Multan, Khanewal), Rechna (Faisalabad, Toba Tek Singh, Jhang), and Chaj (Sargodha) doabs have surpassed safe yields due to insufficient recharge in comparison with abstraction. Due to climatic Pakistan is receiving recurrent flooding events. Other than devastating impacts, these flooding events help to naturally replenish the aquifer, which increases resilience against droughts. Being a sandy aquifer, the flooding events of 2010 and 2014 has played a major role in facilitating groundwater recharge, which is also evident from Figure 3. Due to sufficient recharge, there is not any significant change in an overall increase in area under depth >600 cm covering the period 2010 and 2014 in comparison with 2008 and 2012.

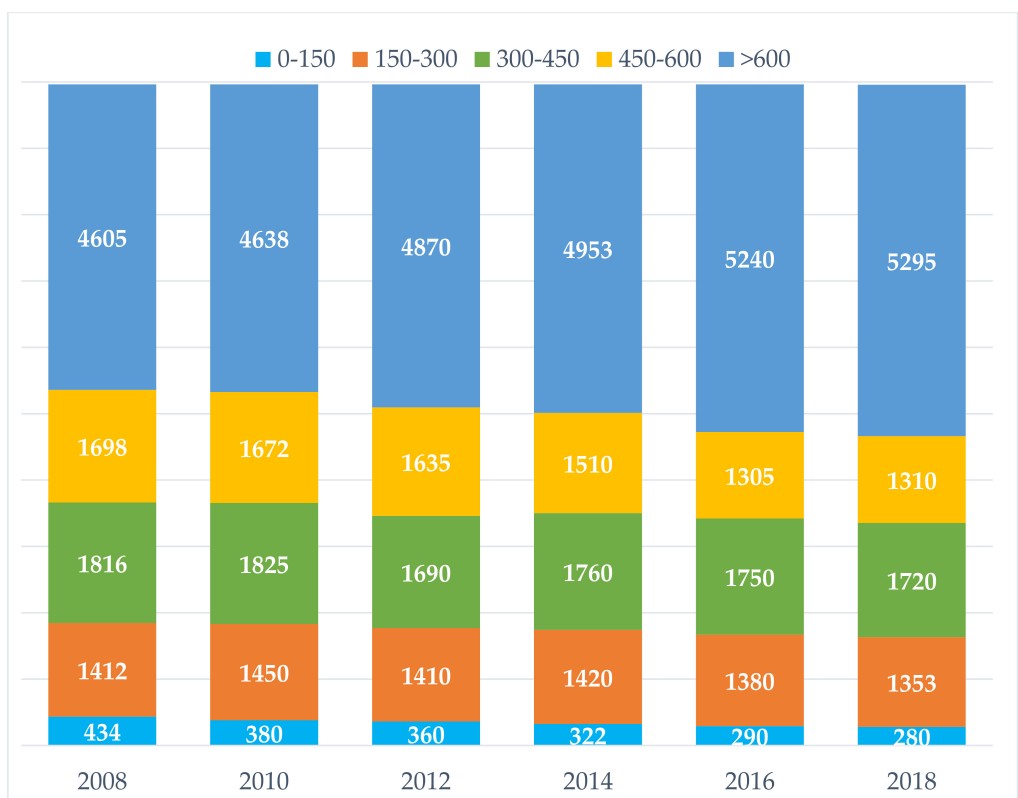

**Figure 3.** Temporal changes in groundwater table depths in Punjab (numbers in 000 ha, DTW range in cm, adopted from [33].

## 3. Data and Methods

The GRACE-based gravity anomaly data of Centre for Space Research (CSR), The University of Texas at Austin-USA RL05 data product from January 2016 to July 2016, is used in this study (available at ftp://podaac.jpl.nasa.gov/allData/grace/L2/CSR/RL05/ (accessed on 15 May 2020). The GRACE mission is a joint venture of National Aeronautics and Space Administration (NASA), USA, and German Aerospace Centre (DLR), Germany, and was launched in 2002. GRACE is a coarse resolution (~160,000 km$^2$) twin gravity measurement satellite mission with global coverage [18]. It collects gravity anomalies through variations in distance between two satellites. While collecting the data, the mass variations on the earth affect the speed of both satellites that changes their distance. When the leading satellite passes over the area having more mass (denser), its speed decreases, whereas the speed of the following satellite increases due to Newton's law of gravitation. The gravity changes constantly due to mass variations over the surface of the earth. These mass changes are mainly due to the changes in either ice or water. Both ice and water frequently change monthly. GRACE is sensitive enough to track these changes from space at an altitude of 480 km. It provides ten daily to monthly gravity anomalies. There are three centers (CSR, JPL, and GFZ) that process the raw data based on different approaches and provide gravity anomalies at different processing levels (shown in Figure 4).

The RL05 is a level-2 GRACE data product that is released after initial filtering. The data filtering and smoothing techniques are applied [29] to infer the Total Water Storage (TWS) from gravity anomalies and then scaling factor is applied for signal restoration [35]. This study uses the methodology defined by [29] and [7] for the extraction of groundwater storage (GWS) changes inferred from GRACE-TWS over the Indus Basin. The Variable Infiltration Capacity (VIC) model-based simulations for soil moisture and runoff fluxes are used to estimate GWS from GRACE-TWS [29,36,37] Figure 5 explains the detailed methodology.

The synthesis of calibration results produced by [29] indicates a good agreement between GRACE-GWS with piezometric data over the Upper Indus Plain. While coming down to the level of each doab, GRACE performed very well for Bari (correlation = 0.93) and Rechna (correlation = 0.65) doabs [7]. For calibration purposes, the piezometric depth to water table data (DTW) has been used to estimate the GWS changes by multiplying the groundwater level changes with specific yield [22,29] These data sets are biannually provided by Scarp Monitoring Organization (SMO-WAPDA). After accuracy evaluation, the current study uses the GRACE inferred GWS anomalies from January to July 2016 for the analysis of aquifer behavior and water use impacts.

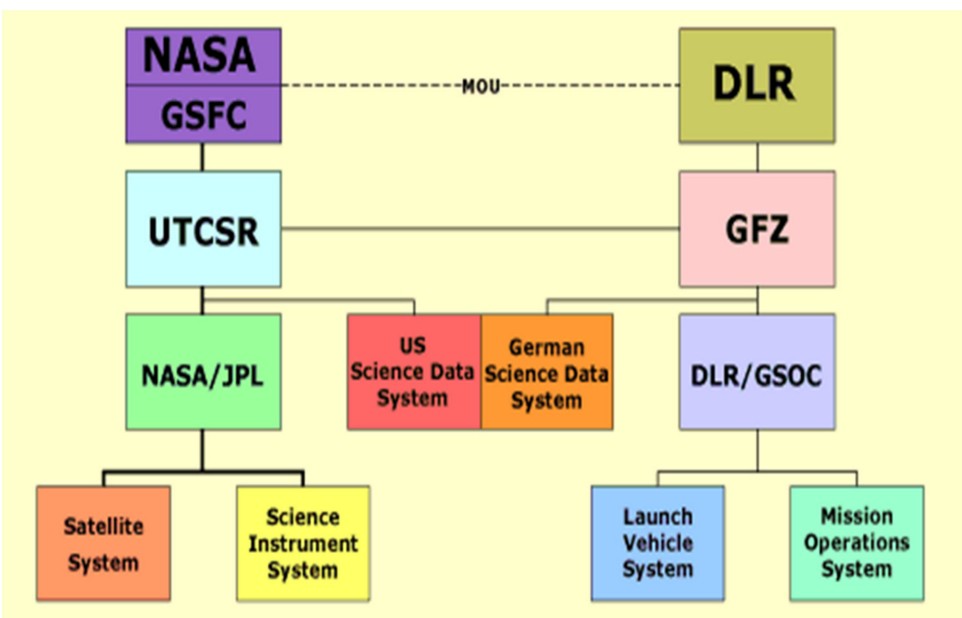

**Figure 4.** GRACE data processing and sharing mechanism. Reproduced from http://www.csr.utexas.edu/grace/mission/project.html (accessed on 15 May 2020).

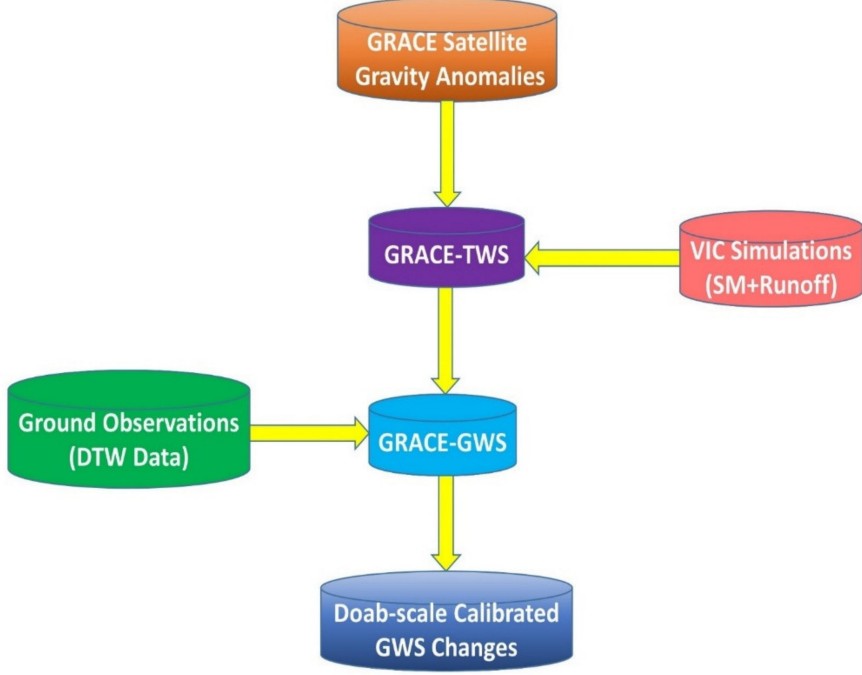

**Figure 5.** Flow chart of the methodology.

## 4. Results and Discussion

The changes in groundwater storage are a function of recharge and pumping [7], which means that changes in groundwater storage are correlated with the variations in recharge and pumping processes, which are an integral component of water balance. Therefore, any change in either groundwater recharge or pumping has a direct impact on groundwater storage [38]. In the Indus Basin, the groundwater system is predominantly influenced by the seasonal fluctuations in the water table due to the monsoon system. The monsoon system is one of the major wind systems of the world and an important source of precipitation in South Asia. The wind blows about six months from the northeast and the remaining six months from the southwest direction. These winds cause seasonal change corresponding to the changes in precipitation, bringing moisture either from the Bay of Bengal or the Arabian Sea. The monsoon system starts at the end of July. So, Scarp Monitoring Organization (SMO) collects depth to water table (DTW) data in a bi-annual mode (pre-monsoon and post-monsoon periods). During the pre-monsoon period, the aquifer undergoes depletion due to excessive pumping for irrigation requirements mainly for rice crops. The rice is traditionally grown through a flood irrigation method by maintaining 3–4 inches of standing water in the field, which is even more than its requirements. However, the rainfall-induced recharge from monsoon events helps to replenish the groundwater system. The recharge depends upon the spatial variations in rainfall along with subsurface lithology over the study area. The analysis of the spatial distribution of rainfall demonstrates that the Rechna and Chaj doabs receive more rainfall than the other two [32]. The analysis of GWS from January through July 2016 reflects an overall depletion trend in groundwater storage captured by GRACE in all four doabs. The maximum depletion is observed during July 2016, referring to the peak pumping conditions over the study period. These negative anomalies are more intense over Rechna, Chaj, and Bari doabs (Figure 6).

The upper parts of Rechna and lower parts of Bari and Chaj doabs show that the aquifer is under stress due to an imbalance between recharge and pumping. The areas of Upper Rechna (Narowal, Gujranwala, and Hafizabad: Figure 6) and Mandi Bahauddin (Chaj doab, Figure 6) belong to the famous rice belt, which requires a lot of water to grow due to prevailing flood irrigation practice in Pakistan. Particularly in Rechna doab, the farmers grow rice a bit earlier and the same impact is quite visible in July 2016.

In the lower Bari doab (Figure 6), the areas of Multan, Khanewal, and Lodhran are under severe groundwater depletion as well as mining conditions [39] with a combined average annual depletion rate of 0.26 m, making a total depletion of about 2.9 m over the period from 2005 to 2015 (Figure 7). Due to the low availability of about 200 mm average rainfall [32] and low flows in rivers Sutlej and Ravi, the pumping dominates the recharge causing water deficit in the form of groundwater mining conditions. Due to the urbanization effect and low flows in River Ravi, the Lahore aquifer (Bari Doab, Figure 6) is also under depletion as reported by GRACE and validated with piezometric data [39]. In the Thal doab (Figure 6), the lower areas are under positive anomalies indicating increment in groundwater storage due to more recharge than pumping through surrounding rivers. In this area, the water table is shallow, and some areas are under physical waterlogging conditions. Figure 8 shows the overall groundwater stock changes with a linear depletion trend whereas Table 1 summarizes the monthly variations in groundwater storage.

The severe depletion in the lower part (Sargodha district) of the Chaj doab (Figure 6) is attributed to increased salinity in groundwater with less fresh groundwater availability putting more pressure in the form of increased pumping [28]. Furthermore, the subsurface lithology having clay lenses restricts the groundwater recharge and the pumping dominates the recharge causing a deficit in groundwater storage.

It is also important to note that there is a visible difference in the scenario of DTW changes before and after 2010 (Figure 7). The scenario before 2010 reflects an increasing trend of DTW, which indicates the impact of extensive pumping in the form of groundwater depletion. This extensive pumping is the result of an increase in tube-well growth as

reflected in Figure 2. However, there was a major flooding event in the study region during late July 2010. This flooding event has facilitated groundwater recharge and somehow replenished the aquifer system. Resultantly, it has changed the pace of groundwater depletion, which seems static to moderate. Rapid population growth and the expansion of agricultural lands have mounted increasing pressure on groundwater resources, resulting in their overexploitation leading to larger drawdown, which leading to groundwater decline as well as a depleting the aquifer.

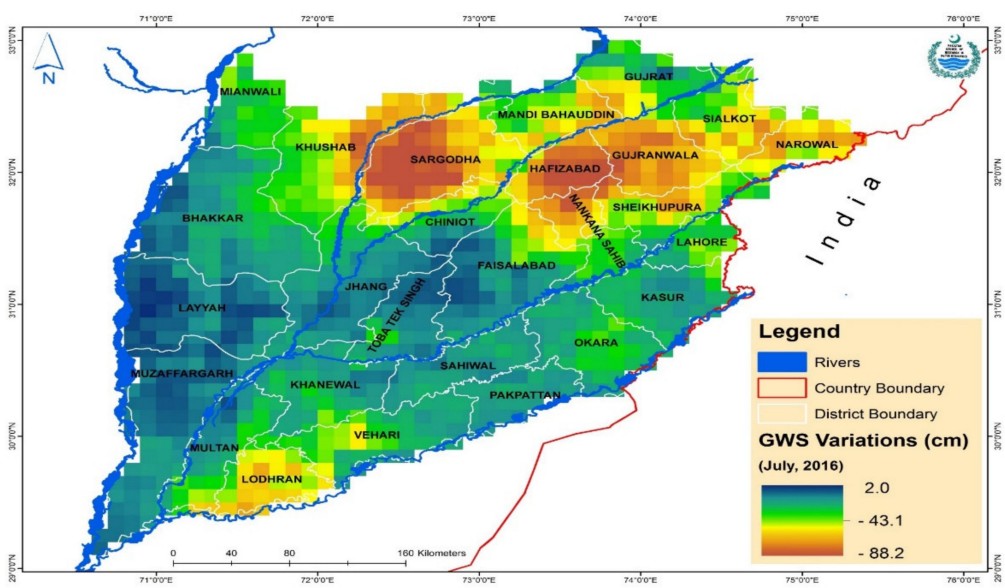

**Figure 6.** Spatial trends in groundwater storage variations during July 2016 over UIPA. Yellow to dark brown are areas under high magnitude negative anomalies, indicating high depletion due to extensive pumping. Blue shows areas with groundwater recharge.

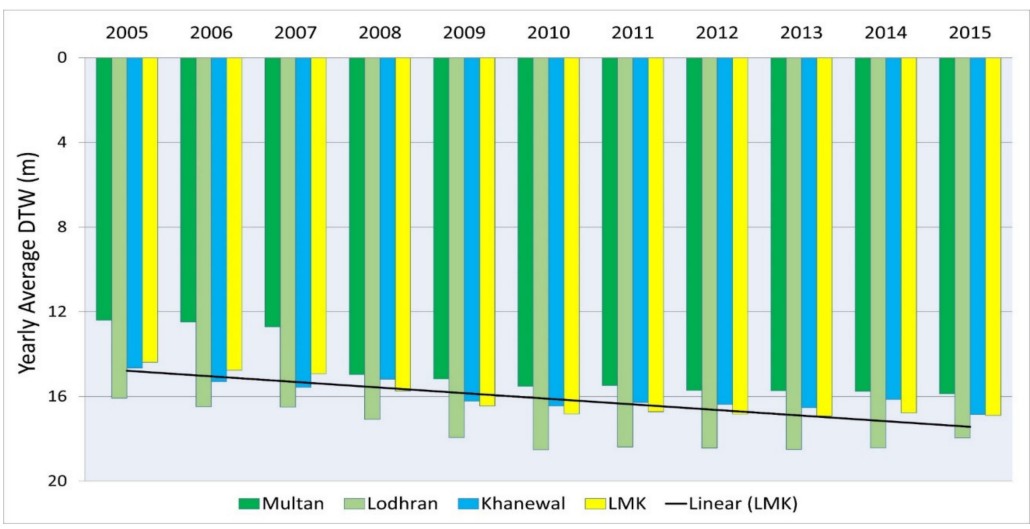

**Figure 7.** Variations in depth to water table (m) from 2005 to 2015 in lower parts of Bari doab.

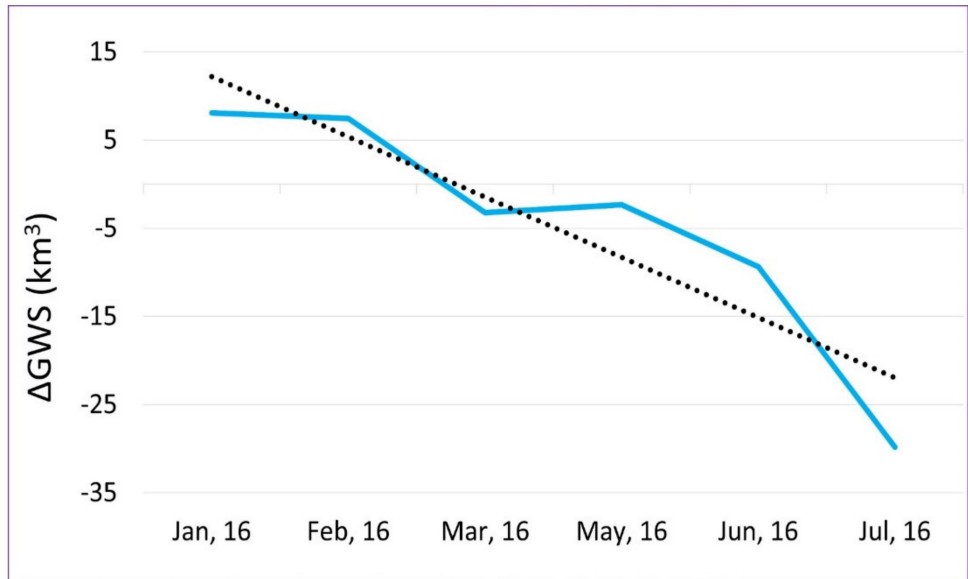

**Figure 8.** Overall picture of groundwater stock from January to July 2016 over UIPA.

**Table 1.** Groundwater storage variations over the Upper Indus Plain.

| Months | ΔGWS (km³) |
| --- | --- |
| January 2016 | 8.08 |
| February 2016 | 7.45 |
| March 2016 | −3.22 |
| May 2016 | −2.32 |
| June 2016 | −9.37 |
| July 2016 | −29.85 |
| GWS Loss (km³) | 1.03 |

## 5. Opportunities and Challenges

The biggest advantage of remote sensing techniques such as GRACE is the global coverage with free data availability. The monthly temporal frequency is also appropriate for basin-scale hydrological applications. It is a marvelous opportunity that GRACE has offered for the first time to support the global aquifer evaluations efforts by overcoming the data gaps. The second feature is its capability to measure and monitor the complete water cycle, which is very difficult otherwise. Thirdly, the role of satellite gravimetry and altimetry is very important in supporting transboundary hydrological studies, particularly in developing the countries' scenario, where the upstream countries are not willing to share accurate and timely information. Consequently, the downstream countries suffer in the form of extreme conditions such as droughts and devastating flooding events. Fourthly, the integration of satellite products with hydrological models potentially supplements each other for any missing information for annual basin-scale budgeting. Most importantly, the monthly scale tracking of groundwater storage changes provides a useful inside of the aquifer system, spatial patterns of groundwater abstraction and identification of critical areas. This information is particularly important and useful for groundwater managers and policymakers to devise appropriate management strategies and implementation of relevant interventions required for the sustainability of this precious groundwater resource. GRACE-derived solutions in the form of total water storage (water height) are also available in the public domain. The purpose of these ready-made solutions is to empower the end-user community for the utility of this TWS information in their applications without bothering the complex processing steps.

Along with opportunities, there are a few challenges too, which limit the societal benefits of remote sensing technology at operational management scales. The inherent limitation of coarse spatial resolution of GRACE satellite hampers the accuracy of results in the study areas less than comparable to its spatial resolution [29]. It is always a trade-off between the accuracy and size of the study area.

The data latency is another hurdle for the timely utilization of GRACE deriver groundwater storage information. The GRACE data processing centers (CSR, JPL, GFZ) take about 1–2 months to publicly release the monthly gravity anomalies in the form of spherical harmonic coefficients after initial processing. In this way, the data frequency is monthly, but it does not become available instantly. It is another challenge yet to overcome by the scientific community to make this satellite-based groundwater storage information readily available to the user community.

The technical complexities and non-availability of related software and programming tools in the public domain further limit the potential of GRACE data in terms of its large-scale operational adaptability. The organizational scale adaptability requires independent GRACE data processing and interpretation capability, which is still a challenging area. The capacity-building efforts and provision of relevant tools in the public domain are required to maximize the potential of GRACE applications for societal benefits.

## 6. Conclusions

This study provides an insight into the satellite remote sensing technological development for groundwater management and uses the case of the Indus Basin to demonstrate its practical adaptability as a tool for groundwater monitoring. Satellite remote sensing is helpful for the assessment of recharge and pumping components, which are an integral part of the formulation of groundwater budgeting at appropriate operational scales for effective groundwater resource management. In the Indus Basin, the unregulated groundwater exploitation has resulted in an excessive lowering of groundwater levels after floods in 2010, whereas before the year 2010, the groundwater depletion rate was nearly static to moderate. Floods played a role as a major recharge source in the year 2010 and replenished the aquifer consisting of sand. Every year, groundwater depletion is dependent on the monsoon, which helps to replenish the aquifer after excessive pumping during the pre-monsoon period. Using GRACE, it is observed that the DTW in the Indus Basin is increasing at an alarming rate. Furthermore, it is shown that GRACE proves to be a good research tool if the recharge or pumping phenomenon in the study area (even at small operation scales) is consistent and significant enough, though it may not be effective for the intermixed phenomenon at scales not comparable to its actual resolution. Based on the assessment provided in this study, it is recommended that the impact of climatic variables on groundwater storage should be further explored.

**Author Contributions:** Conceptualization, validation, writing—original draft, G.A.; funding acquisition, project administration, resources, Y.G.; software, investigation, writing review and editing, N.I.; data curation, methodology, Y.S.; visualization, formal analysis, M.H. All authors have read and agreed to the published version of the manuscript.

**Funding:** This research is financially supported by the Chinese Academy of Sciences President's International Fellowship Initiative (Grant No. 2021VCA0010) and the International Science & Technology Cooperation Program of China (2018YFE0100100).

**Data Availability Statement:** Publicly available datasets were analyzed in this study. This data can be found here https://irrigation.punjab.gov.pk/ and ftp://podaac.jpl.nasa.gov/allData/grace/L2/CSR/RL05/.

**Acknowledgments:** This research is financially supported by the Chinese Academy of Sciences President's International Fellowship Initiative (Grant No. 2021VCA0010) and the International Science & Technology Cooperation Program of China (2018YFE0100100). The authors would like to greatly acknowledge the support extended by Quaid-i-Azam University and the Pakistan Council of

Research in Water Resources to accomplish this study. The data contribution of Punjab Irrigation Department, Lahore, and SMO, Lahore is also acknowledged.

**Conflicts of Interest:** The authors declare no conflict of interest.

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
