# Peer review of "Appraisal of Remote Sensing Technology for Groundwater Resource Management Perspective in Indus Basin"

_sustainability, doi:10.3390/su13179686_

Round 1

Reviewer 1 Report

This study investigated remote sensing technology for groundwater resource management in Indus Basin.  The topic of the paper is attractive. But there is no deep analysis. I recommend the article for publication after making major revision:

Major Concerns.

  1. figure 1 study area should include country boundary and district. 
  2. the font sizes of the figures are too small and the readers may hard to capture the information. 
  3. I suggest the authors make more analysis on the relationship between tube-wells growth and yearly average DTW. For example, before 2010, yearly average DTW show increase trend, however, after 2010, it seems there is no increase trend. Yet, Tube-wells growth rapidly from 2006-2016. What makes them different?

Reviewer 2 Report

The paper titled "APPRAISAL OF REMOTE SENSING TECHNOLOGY FOR GROUNDWATER RESOURCE MANAGEMENT PERSPECTIVE IN INDUS BASIN" is an important topic to the study region.

General Comments

The Manuscript needs more Literature review about the topic in the region area

The results needs more information and more explanation

The conclusion needs rewritten and more results to be added

 Other Comments

Introduction:

page 2 (last paragraph): the sentence of "The solutions demand the integration of interdisciplinary techniques." needs more rewriting and more clarification

The introduction needs some works and literature review about the topic in the region area

The objective of the study as it stated is not enough for the paper and the work is not only what it is written, "The objective of this study is to review the current practices, available sources of information, and the gap in analysis from the perspective of groundwater resource monitoring"

Another title is needed before " 2. Upper Indus Plain Aquifer" like study area

Rewrite in page 4: In response, farmers are growing more crops at the cost of pumping more and more groundwater

Page 7  Results: The sentence of " The changes in groundwater storage are a function of recharge and pumping." needs reference

Please define the monsoon system mentioned in the study (Page 7)

Figure 6 : There is no units for the water table

How can you explain the negative results in Table 1

In the conclusion ". This study also outlines the future direction of research to further explore the relationship of climatic variables with groundwater storage changes and simulation of future projected scenarios for the impacts of climatic implications on water resources." there is no evidence for that in the text

The paper titled "APPRAISAL OF REMOTE SENSING TECHNOLOGY FOR GROUNDWATER RESOURCE MANAGEMENT PERSPECTIVE IN INDUS BASIN" is an important topic to the study region.

General Comments

The Manuscript needs more Literature review about the topic in the region area

The results needs more information and more explanation

The conclusion needs rewritten and more results to be added

 Other Comments

Introduction:

page 2 (last paragraph): the sentence of "The solutions demand the integration of interdisciplinary techniques." needs more rewriting and more clarification

The introduction needs some works and literature review about the topic in the region area

The objective of the study as it stated is not enough for the paper and the work is not only what it is written, "The objective of this study is to review the current practices, available sources of information, and the gap in analysis from the perspective of groundwater resource monitoring"

Another title is needed before " 2. Upper Indus Plain Aquifer" like study area

Rewrite in page 4: In response, farmers are growing more crops at the cost of pumping more and more groundwater

Page 7  Results: The sentence of " The changes in groundwater storage are a function of recharge and pumping." needs reference

Please define the monsoon system mentioned in the study (Page 7)

Figure 6 : There is no units for the water table

How can you explain the negative results in Table 1

In the conclusion ". This study also outlines the future direction of research to further explore the relationship of climatic variables with groundwater storage changes and simulation of future projected scenarios for the impacts of climatic implications on water resources." there is no evidence for that in the text

The paper titled "APPRAISAL OF REMOTE SENSING TECHNOLOGY FOR GROUNDWATER RESOURCE MANAGEMENT PERSPECTIVE IN INDUS BASIN" is an important topic to the study region.

General Comments

The Manuscript needs more Literature review about the topic in the region area

The results needs more information and more explanation

The conclusion needs rewritten and more results to be added

 Other Comments

Introduction:

page 2 (last paragraph): the sentence of "The solutions demand the integration of interdisciplinary techniques." needs more rewriting and more clarification

The introduction needs some works and literature review about the topic in the region area

The objective of the study as it stated is not enough for the paper and the work is not only what it is written, "The objective of this study is to review the current practices, available sources of information, and the gap in analysis from the perspective of groundwater resource monitoring"

Another title is needed before " 2. Upper Indus Plain Aquifer" like study area

Rewrite in page 4: In response, farmers are growing more crops at the cost of pumping more and more groundwater

Page 7  Results: The sentence of " The changes in groundwater storage are a function of recharge and pumping." needs reference

Please define the monsoon system mentioned in the study (Page 7)

Figure 6 : There is no units for the water table

How can you explain the negative results in Table 1

In the conclusion ". This study also outlines the future direction of research to further explore the relationship of climatic variables with groundwater storage changes and simulation of future projected scenarios for the impacts of climatic implications on water resources." there is no evidence for that in the text

The paper titled "APPRAISAL OF REMOTE SENSING TECHNOLOGY FOR GROUNDWATER RESOURCE MANAGEMENT PERSPECTIVE IN INDUS BASIN" is an important topic to the study region.

General Comments

The Manuscript needs more Literature review about the topic in the region area

The results needs more information and more explanation

The conclusion needs rewritten and more results to be added

 Other Comments

Introduction:

page 2 (last paragraph): the sentence of "The solutions demand the integration of interdisciplinary techniques." needs more rewriting and more clarification

The introduction needs some works and literature review about the topic in the region area

The objective of the study as it stated is not enough for the paper and the work is not only what it is written, "The objective of this study is to review the current practices, available sources of information, and the gap in analysis from the perspective of groundwater resource monitoring"

Another title is needed before " 2. Upper Indus Plain Aquifer" like study area

Rewrite in page 4: In response, farmers are growing more crops at the cost of pumping more and more groundwater

Page 7  Results: The sentence of " The changes in groundwater storage are a function of recharge and pumping." needs reference

Please define the monsoon system mentioned in the study (Page 7)

Figure 6 : There is no units for the water table

How can you explain the negative results in Table 1

In the conclusion ". This study also outlines the future direction of research to further explore the relationship of climatic variables with groundwater storage changes and simulation of future projected scenarios for the impacts of climatic implications on water resources." there is no evidence for that in the text

The paper titled "APPRAISAL OF REMOTE SENSING TECHNOLOGY FOR GROUNDWATER RESOURCE MANAGEMENT PERSPECTIVE IN INDUS BASIN" is an important topic to the study region.

General Comments

The Manuscript needs more Literature review about the topic in the region area

The results needs more information and more explanation

The conclusion needs rewritten and more results to be added

 Other Comments

Introduction:

page 2 (last paragraph): the sentence of "The solutions demand the integration of interdisciplinary techniques." needs more rewriting and more clarification

The introduction needs some works and literature review about the topic in the region area

The objective of the study as it stated is not enough for the paper and the work is not only what it is written, "The objective of this study is to review the current practices, available sources of information, and the gap in analysis from the perspective of groundwater resource monitoring"

Another title is needed before " 2. Upper Indus Plain Aquifer" like study area

Rewrite in page 4: In response, farmers are growing more crops at the cost of pumping more and more groundwater

Page 7  Results: The sentence of " The changes in groundwater storage are a function of recharge and pumping." needs reference

Please define the monsoon system mentioned in the study (Page 7)

Figure 6 : There is no units for the water table

How can you explain the negative results in Table 1

In the conclusion ". This study also outlines the future direction of research to further explore the relationship of climatic variables with groundwater storage changes and simulation of future projected scenarios for the impacts of climatic implications on water resources." there is no evidence for that in the text

Round 2

Reviewer 1 Report

I have no more comment and suggest the editors that the paper could be accepted in the present form.

Reviewer 2 Report

The authors of the paper  "APPRAISAL OF REMOTE SENSING TECHNOLOGY FOR GROUNDWATER RESOURCE MANAGEMENT PERSPECTIVE IN INDUS BASIN" are improved their Text. There is still points to be added in the results (add more results) and the English language needs more editing.

Other comments:

- A study area section is needed, please rearrange the 2.Upper Indus Plain aquifer to be a study area section.

- Explain the decline trend for the years 2015 and 2016 in Fig. 2

- Explain in the text the relationship between DTW and recharge amounts of the aquifer

Author Response

The authors of the paper  "APPRAISAL OF REMOTE SENSING TECHNOLOGY FOR GROUNDWATER RESOURCE MANAGEMENT PERSPECTIVE IN INDUS BASIN" are improved their Text. There is still points to be added in the results (add more results) and the English language needs more editing.

Addressed the comments in the manuscript as proposed by a reviewer.

Other comments:

 - A study area section is needed, please rearrange the 2. Upper Indus Plain aquifer to be a study area section.

The Upper Indus Plain aquifer is replaced with a study area in the manuscript as suggested by the reviewer.

- Explain the decline trend for the years 2015 and 2016 in Fig. 2

Explained the declining trend for the years 2015 & 2016 in the manuscript.

- Explain in the text the relationship between DTW and recharge amounts of the aquifer

Explained in the manuscript as proposed by a reviewer. 

Round 3

Reviewer 2 Report

The Manuscript is improved and the authors are included all my comments. The Manuscripts needs intensive English edit and the authors have to arrange the titles and References according to the Journal instructions.